# Prevalence, treatment, and outcomes of hepatitis C in an MDR/RR-TB trial cohort

**Jocelyn Jansen van Vuuren**[1�***], **Tim Crocker-Buque**[2�***], **Catherine Berry**[1�***]*,
**Dzmitry Viatushka**[3‡], **Ruzilya Usmanova**[4‡], **Bern-Thomas Nyang'wa**[1], **Nargiza Parpieva**[3],
**Irina Liverko**[3], **Varvara Solodovnikova**[4], **Animesh Sinha**[1]

**1** Médecins sans Frontières, United Kingdom & Netherlands, **2** London School of Hygiene & Tropical Medicine, London, United Kingdom, **3** Belarus Republican Specialized Scientific Practice Medical Center of Phthisiology and Pulmonology, Minsk, Belarus, **4** Republic Scientific and Practical Centre for Pulmonology and Tuberculosis, Tashkent, Uzbekistan

☯ These authors contributed equally to this work.
‡ DV and RU also contributed equally to this work.
* catherine.berry@london.msf.org

**Data Availability Statement:** All data constituting the minimal dataset is available via TB-PACTS (https://c-path.org/tools-platforms/tb-pacts) using

## Abstract

Tuberculosis (TB) and chronic hepatitis C virus infection (HCV) remain significant global health challenges, especially in low- and middle-income countries. In Eastern Europe, a considerable percentage of multi-drug resistant (MDR) and rifampicin resistant (RR) TB populations show high HCV prevalence. Current WHO guidelines do not routinely advise HCV testing during MDR-TB treatment, despite HCV being a risk factor for drug-induced liver complications in TB patients. This study investigates the co-treatment of MDR/RR-TB and HCV, using data from the TB-PRACTECAL trial. Data were collected as part of the TB-PRACTECAL clinical trial. All participants were screened for HCV at baseline. Participants who were HCV antibody positive and those who were treated for hepatitis C with Direct Acting Antivirals (DAAs) were extracted and compared to overall cohort characteristics. The characteristics of participants concomitantly treated with direct-acting antivirals are described including hepatitis treatment outcomes and adverse events. Among 552 participants from Belarus, Uzbekistan, and South Africa, 24 (4.3%) were HCV antibody positive. Unfavourable TB treatment outcomes were noted in 106/523 (22%) of the HCV-negative, 8/18 (44%) of the HCV-seropositive, and 2/7 (29%) of HCV-confirmed participants treated with DAAs. Of the six participants who received concurrent HCV and MDR/RR TB treatment, three were cured of HCV and three had no post-treatment HCV RNA test, five completed TB treatment and one discontinued treatment due to a severe adverse reaction. Concurrent treatment of MDR-TB and HCV, including in HIV patients, showed promising outcomes with no significant adverse events. The findings support the potential benefits of integrating HCV care into MDR-TB management.

## Introduction

Tuberculosis (TB) and chronic hepatitis C virus infection (HCV) are both major global public health problems and disproportionately affect persons in low- and middle-income countries

study reference TB1035. Data sets hosted on TB-PACTS are made freely available to researchers following a simple application process. Once approved, researchers can access patient-level de-identified data from TB-PRACTECAL. This has been the preferred method agreed upon by the sponsor, Medicèns Sans Frontierés in consultation with data protection advisors considering the individualised nature of TB care. Researchers would be able to access these data in the same manner as the authors and the authors do not have any special access or request privileges that others do not have.

**Funding:** This work was supported by Médecins Sans Frontières (MSF), who financially supported the submission fee to the journal. The TB-PRACTECAL trial from where the main data was used was funded by Médecins sans Frontières (MSF) (ClinicalTrials.gov number: NCT02589782). C.B., B-T.N., A.S. report employment with MSF. JJvV reports previous employment with MSF but did not receive any grants towards this work.The funders had no role in study design, analysis, decision to publish, or preparation of the manuscript. The funders provided data for collection and analysis from TB-PRACTECAL trial.

**Competing interests:** I have read the journal's policy and the authors of this manuscript have the following competing interests: C. B. reports a role as ACTnet board member (research network). No other conflicts reported.

(LMICs) [1, 2]. The prevalence of HCV is high among people with active TB, and up to 30% in multi-drug resistant (MDR) and rifampicin-resistant (RR) TB populations, particularly in Eastern Europe [3–6]. Routine, systematic testing for HCV is not currently recommended in World Health Organization (WHO) guidelines for MDR-TB treatment [7]. Only one in five people living with hepatitis C know their status, around two-thirds of those diagnosed receive treatment, and this disproportionately affects those who are economically disadvantaged, displaced, migrants, and rural populations [8]. HCV is an independent risk factor for hepatotoxicity and drug-induced liver injury (DILI) in people with TB due to the risk of underlying chronic liver damage [9–15]. There is no clear recommendation on co administering Direct Acting Antivirals (DAAs) with second line anti TB drugs [16].

Some data already suggests that concomitant MDR-RR/TB and HCV management is safe, effective, and feasible. Daclatasvir/sofosbuvir or velpatasvir/sofosbuvir with and without bedaquiline containing second line anti-tubercular regimens was shown to be effective in 18–20 month regimens [17–20]. However, the participants from these studies were treated prior to the World Health Organisation update on MDR-TB management in 2022 [7]. It is also currently unknown whether HCV viral suppression could improve tolerance and outcomes of potentially hepatotoxic TB treatment regimens. The WHO now recommends a new 6 month all oral regimen, including bedaquiline, pretomanid, linezolid with and without moxifloxacin for MDR/RR-TB using data from TB-PRACTECAL trial [21] and has put out a call for expertise on the co-administration of treatment for HCV and RR/MDR-TB [22].

For this report, we aimed to examine the outcomes of HCV seropositive participants, and participants who had confirmatory tests and were concomitantly treated within the trial. We aimed to describe the HCV serostatus within the trial cohort, report on TB and HCV outcomes by HCV serostatus, evaluate differences in liver function between HCV seropositive patients, HCV seronegative patients, and HCV seropositive patients treated with DAAs during TB treatment, and finally, describe the treatment course, adverse events, HCV and TB outcomes of the cohort of patients that were concomitantly treated.

## Methods

This study presents a secondary analysis of data collected as part of the TB-PRACTECAL clinical trial [21, 23]. Participants with confirmed MDR/RR-TB were included in the trial. Inclusion criteria included age 15 years and above and confirmed resistance to at least rifampicin via molecular or phenotypic microbiological testing. Patients received bedaquiline, pretomanid, and linezolid (BPaL) based regimens, with or without moxifloxacin (BPaLM) or clofazimine (BPaLC) for the treatment of RR/MDR TB or the locally accepted standard of care in line with WHO guidelines in place at the time. Participants were excluded if their baseline ALT or AST were more than 3 times the upper limit of normal (ULN). All participants living with HIV included irrespective of CD4 count and offered anti-retroviral therapy as part of the trial protocol.

For this analysis, all participants who were screened at baseline for HCV antibodies using a validated immunoassay were included. Participants were classified as HCV antibody negative (ABNeg), HCV antibody positive (ABPos), and HCV antibody positive and confirmed viraemia on HCV polymerase chain reaction (PCR) or HCV viral load (viral load) and receiving DAAs during treatment. Participants were considered to be HCV antibody positive if they tested positive on antibody testing or was reported as past medical history. Concomitant treatment data were used to identify participants prescribed DAAs during the trial. Confirmatory HCV PCR testing was not required as part of core data, test access varied by site within their local framework and was not available to participants, but was a pre-requisite for treatment. Treatment of HCV was at the discretion of the investigator and subject to national guidelines.

Included variables were demographics, co-morbidities (hepatitis B and HIV serostatus), and social history (alcohol and drug use), TB treatment drugs, biochemical tests (haematological and liver function test results), adverse event reports, clinical trial outcomes, and, where applicable, HCV treatment outcomes. Outcome definitions used in this paper reflect the TB-PRACTECAL trial definitions [23]. The primary TB treatment outcome measure was a composite of unfavourable outcome at 72 weeks, including death, loss to follow-up (LTFU), early discontinuation (ED), treatment failure (TF), and TB recurrence.

Safety was monitored at every visit by clinical history, physical examination, and biochemistry (including complete blood count, liver function tests, urea and electrolytes, and lipase), and other testing in line with the investigational schedule. Adverse events were graded on entry into the clinical trials database, using thresholds described in the study protocol.

We analysed the results of biochemical and haematological tests as continuous variables using all available results at the specified timepoints to calculate mean results. To report the adverse events recorded for the DAA cohort, we grouped the data into three categories: haematological (anaemia, lymphopenia, neutropenia), hepatic (elevation in alanine aminotransferase (ALT), aspartate aminotransferase (AST), alkaline phosphatase (ALP), and bilirubin) and gastrointestinal (nausea, abdominal pain, vomiting, diarrhoea). The onset of an adverse event is shown in relation to TB and DAA treatment commencement.

The descriptive characteristics and safety outcomes used the as-treated safety population (those who were randomised and dispensed study medication on at least one occasion). For treatment outcomes, we used the modified intention to treat (mITT) population (the as-treated population excluding participants who had unconfirmed MDR/RR-TB (i.e participants found to be sputum-culture negative, or where rifampicin-resistance was discordant on confirmatory testing). The mITT group was used in TB-PRACTECAL to determine efficacy results.

Knime Analytics Platform (version 4.6.5; Knime AG, Zurich, Switzerland) and R Studio (version 4.3.1) was used to enter raw data obtained from the trial and do the descriptive analysis of the prevalence of hepatitis C seropositivity, effects on tuberculosis treatment outcomes and compared the liver function of the three groups over time. We use box plots to display the differences between liver function of the groups. We used Microsoft Excel (version 2307, Microsoft 365) to create a demographic table of the concomitantly treated participants including outcomes of both infections, and figures displaying their progress in the trial.

Ethical approval: this study was a post-hoc analysis of the TB-PRACTECAL clinical trial data set and complemented by routinely collected data. It was considered exempt from ethical review.

See S1 Checklist for the STROBE checklist.

## Results

Participants were enrolled from Belarus, Uzbekistan, and South Africa from January 2017 to March 2021. Fig 1 shows the participants included in this analysis.

Of 552 participants randomised, all were screened for HCV and 24 (4%) were HCV seropositive. Excluding two participants who had invalid HCV antibody results, the as-treated cohort was 548 and the mITT outcomes cohort included 505 participants which were grouped by HCV status. One participant was seronegative at baseline but due to high index of suspicion received repeat testing locally and was later found to be HCV PCR positive. This participant was included in the concomitantly treated cohort (DAA). Demographics and trial outcome data are presented in **Table 1**.

Participants in the ABPos and DAA groups were older (median 44 years, compared to 35) with a higher proportion of males than the ABNeg group (67% and 86%, compared to 59%).

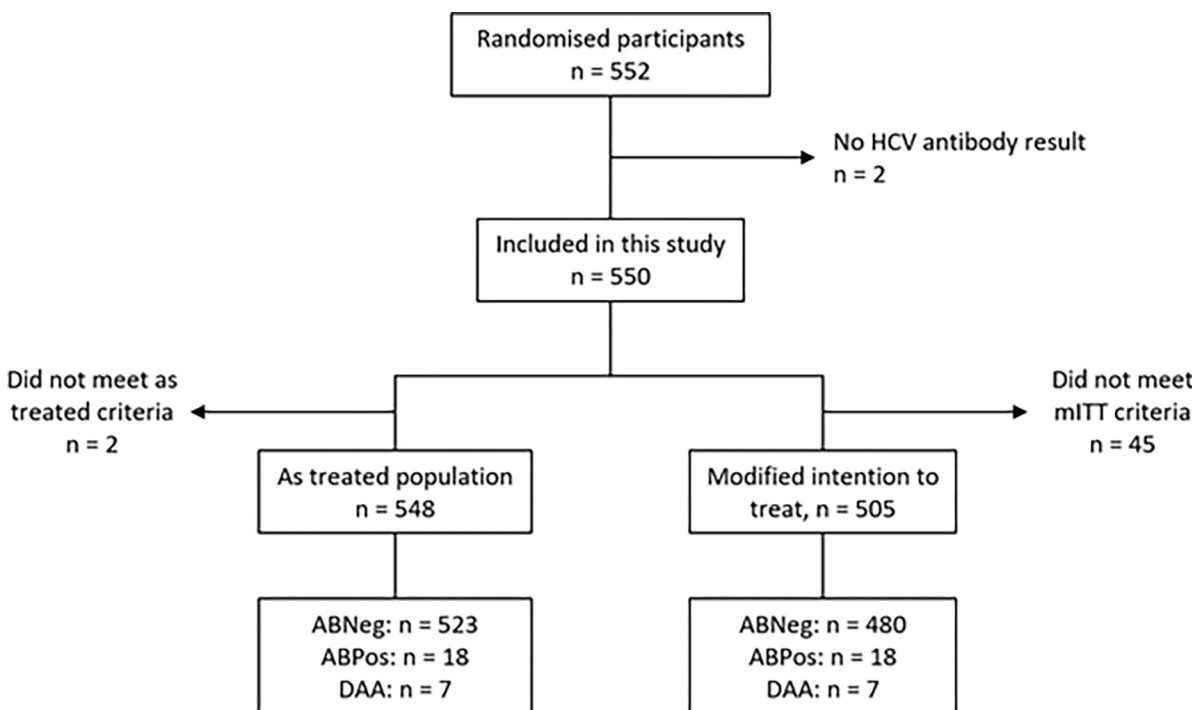

**Fig 1. Showing the flow of included and excluded participants in this analysis.** mITT: modified intention to treat = as treated population, ABNeg: hepatitis C antibody negative, ABPos: hepatitis C antibody positive, DAA: hepatitis C antibody positive and treated with direct acting antivirals.

Hepatitis B co-infection was highest in the ABPos group (17%) compared to the ABNeg (2%) and none in the DAA group. HIV co-infection was highest in the DAA group (71%) compared to the ABPos (39%) and ABNeg (27%) groups. Limited data were available on intravenous drug use and alcohol status, but in the DAA group 2 participants had a history of injecting drug use and 3 reported excess alcohol use. In the DAA group, 6 of 7 participants received a trial regimen containing bedaquiline (Table 2). Mean ALT in the DAA group measured at screening was slightly higher (36 IU/L) and above the upper limit of normal range (35IU/L) than in the ABPos and ABNeg groups (29 and 28 IU/L). 86% of DAA participants had ALT above normal, compared to 50% of the ABPos and 26% of the ABNeg groups. Unfavourable outcomes in the mITT cohort were highest in the ABPos group (44%) and lower in the ABNeg (22%) and DAA group (29%). The contributing negative outcomes were early discontinuation in the ABPos and DAA groups (33% and 29% respectively), with no deaths recorded in either.

Table 2 shows the characteristics of the 6 participants in the DAA group who received a bedaquiline containing regimen at the same time as receiving DAA treatment alongside their demographic characteristics, co-morbidities, HCV RNA status and treatment outcomes. DAA treatment in all participants was with a combination of daclatasvir and sofosbuvir.

The 7[th] participant did not receive treatment concomitantly and so is not included in the nested case series. Fig 2 shows the 6 participants' journey through the study, highlighting the timing of both treatment regimens and adverse events in the following categories: haematological, hepatic, and gastrointestinal. Five of the 6 concurrently treated participants completed TB treatment and 1 withdrew consent. Aside from participant 6, the others commenced DAAs between weeks 2 and 5, and had completed concurrent DAAs between weeks 14 and 25. Participant 6 suffered liver dysfunction related to TB treatment and was later treated with DAAs between weeks 15 and 27, after this had resolved. All completed HCV treatment with DAAs

**Table 1. Demographic and clinical characteristics of the TB PRACTECAL intention to treat cohort and outcomes of the modified intention to treat cohort by hepatitis C antibody status and treatment with direct acting antivirals.** ABNeg: HCV antibody negative, ABPos: HCV antibody positive, DAA: treated with direct acting antivirals, ITT: intention to treat, mITT: modified intention to treat, HCV: hepatitis C virus, HBV: hepatitis B virus, HIV: human immunodeficiency virus, ALT: alanine transaminase, LTFU: lost to follow-up, SOC: standard of care BPaLM: Bedaquiline, Pretomanid, Linezolid, Moxifloxacin, BPaLC: Bedaquiline Pretomanid, Linezolid, Clofazimine, BPaL Bedaquiline Pretomanid, Linezolid. *one participant HCV seronegative at baseline, but retested due to clinical suspicion and treated with DAAs, ED: early discontinuation of treatment, LTFU: Lost to follow up after treatment completion. Country level individual data withhold to preserve patient confidentiality.

**Baseline characteristics (as-treated population, n = 548)**

|  |  | ABNeg | ABPos | DAA |
|---|---|---|---|---|
|  |  | n = 523 | n = 18 | n = 7 |
| Age | Median (range) | 35 (15–73) | 44 (29–57) | 44 (28–51) |
| Sex (N, %) | Male | 307 (59) | 12 (67) | 6 (86) |
| HCV antibody (N, %) | Positive | 0 (0) | 18 (100) | 6 (86)* |
| HBV surface antigen (N, %) | Positive | 36 (2) | 3 (17) | 0 (0) |
| HIV (N, %) | Positive | 141 (27) | 7 (39) | 5 (71) |
| ALT IU/L | Mean | 28 | 29 | 36 |
| ALT (N, %) | >35 IU/L | 138 (26) | 9 (50) | 6 (86) |
| TB regimen (N, %) | SOC | 141 (27) | 7 (39) | 1 (14) |
|  | BPaLM | 144 (28) | 4 (22) | 3 (43) |
|  | BPaL | 117 (22) | 4 (22) | 1 (14) |
|  | BPalC | 121 (23) | 3 (17) | 2 (29) |

**TB treatment outcomes (mITT population, n = 505)**

|  |  | ABNeg | ABPos | DAA |
|---|---|---|---|---|
|  |  | n = 480 | n = 18 | n = 7 |
| 72w outcome (all cause) (N, %) | Unfavourable | 106 (22) | 8 (44) | 2 (29) |
| 72w outcome (N, %) | Death | 7 (2) | 0 (0) | 0 (0) |
|  | Failure | 1 (<1) | 0 (0) | 0 (0) |
|  | Recurrence | 7 (2) | 2 (11) | 0 (0) |
|  | LTFU | 14 (3) | 0 (0) | 0 (0) |
|  | Non assessable | 1 (<1) | 0 (0) | 0 (0) |
|  | ED | 76 (16) | 6 (33) | 2 (29) |
| Early discontinuation reason (N, %) | Adverse event | 38 (8) | 2 (11) | 0 (0) |
|  | Adherence | 18 (4) | 2 (11) | 1 (14) |
|  | Withdrew | 11 (2) | 1 (6) | 1 (14) |
|  | Exclusion | 6 (1) | 0 (0) | 0 (0) |
|  | Other | 3 (<1) | 1 (6) | 0 (0) |

**Table 2. Showing included patients treated for hepatitis C within the TB PRACTECAL cohort with treatment outcomes.** DAA: direct acting antiviral, M: male, F: female, Bdq: Bedaquiline, Cfz: Clofazimine, Lzd: Linezolid, Lfx: Levofloxacin, Cs: Cycloserine, Pa: Pretomanid, Mfx: Moxifloxacin.

| No. | Sex | Age | HIV status | Alcohol Use | Reported IVDU | TB treatment | TB treatment outcome | Baseline HCV Viral load (IU/L) | HCV treatment Outcome | HCV VL post DAA |
|---|---|---|---|---|---|---|---|---|---|---|
| 1 | M | 43 | Positive | Yes | Yes | Bdq, Cfz, Lzd, Lfx, Cs | Discontinued | $3,3 \times 10^6$ | Completed | Unknown |
| 2 | M | 54 | Negative | Yes | Unknown | BPaLM | Completed | $9,88 \times 10^5$ | Completed | Unknown |
| 3 | M | 37 | Negative | No | No | BPaLM | Completed | $1,3 \times 10^6$ | Completed | Undetectable |
| 4 | F | 49 | Positive | No | No | BPaL | Completed | $5,3 \times 10^6$ | Completed | Undetectable |
| 5 | M | 46 | Positive | Unknown | Unknown | BPaLC | Completed | Unknown | Extended | Unknown |
| 6 | M | 49 | Positive | No | No | BPaLM | Completed | $4,88 \times 10^6$ | Completed | Undetectable |

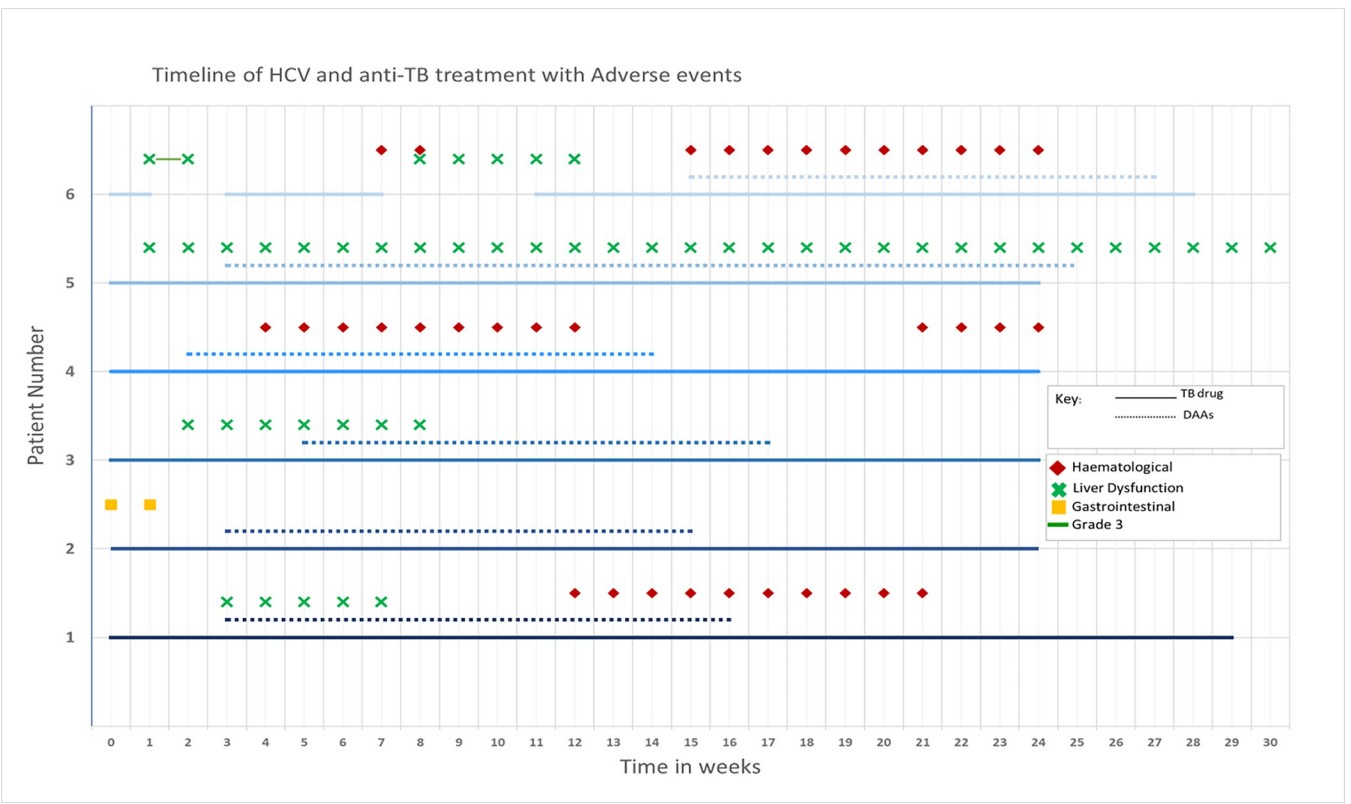

**Fig 2. Showing the timeline of individual patients treated for hepatitis C within the TB PRACTECAL cohort alongside TB treatment and HCV treatment time and duration, as well as haematological, liver and gastrointestinal adverse events.** Haematological AEs: anaemia, leukopenia, lymphopenia, or neutropenia, Liver dysfunction: raised alanine transaminase (ALT), aspartate transaminase (AST) or alkaline phosphatase (ALP). Gastrointestinal Aes: diarrhoea, heartburn, nausea, or vomiting. Grade 3: classified as liver enzymes >5x 20x upper limit of normal.

and 3 had evidence of confirmed cure, HCV viral load measured at 12 weeks post treatment is only known for 3 of the participants as this was not a requirement of the trial and wasn't routinely available at all sites. One participant had HCV treatment extended, although no further information was available to us.

Regarding safety, the only grade 3 adverse event was in participant 6 with liver dysfunction considered related to TB treatment in weeks 1–2, which resolved with cessation and did not recur with re-introduction of medications. Participant 5 had sustained grade 1–2 raise in AST/ALT at enrolment that resolved after completion of all treatment. Participant 3 had grade 1–2 raised liver enzymes at introduction of TB medications, which resolved. Participant 1 had grade 1–2 raised liver enzymes at DAA introduction, which resolved. Participants 1, 4 and 6 exhibited grade 1–2 haematological dysfunction (anaemia) during the period of concomitant DAA treatment, which resolved. The only gastrointestinal side effects were recorded in participant 2 at commencement of TB medications.

Box plots of the median and interquartile range of ALT, ALP and bilirubin in the as-treated cohort as measured at visits 4 (week 1), 6 (week 4), 8 (week 12), 10 (week 20), 12 (week 32) and 17 (week 72) are shown in Figs 3–6. At visit 4 (week 1) ALT was higher in the DAA group and above the ULN than in the ABPos and ABNeg groups, suggesting a degree of underlying liver dysfunction. This difference is not observed from subsequent visits onwards post the initiation of treatment in majority of concomitantly treated patients. Median AST was at the upper limit of normal in the DAA group and slightly above in the ABPos group at week 1. The DAA

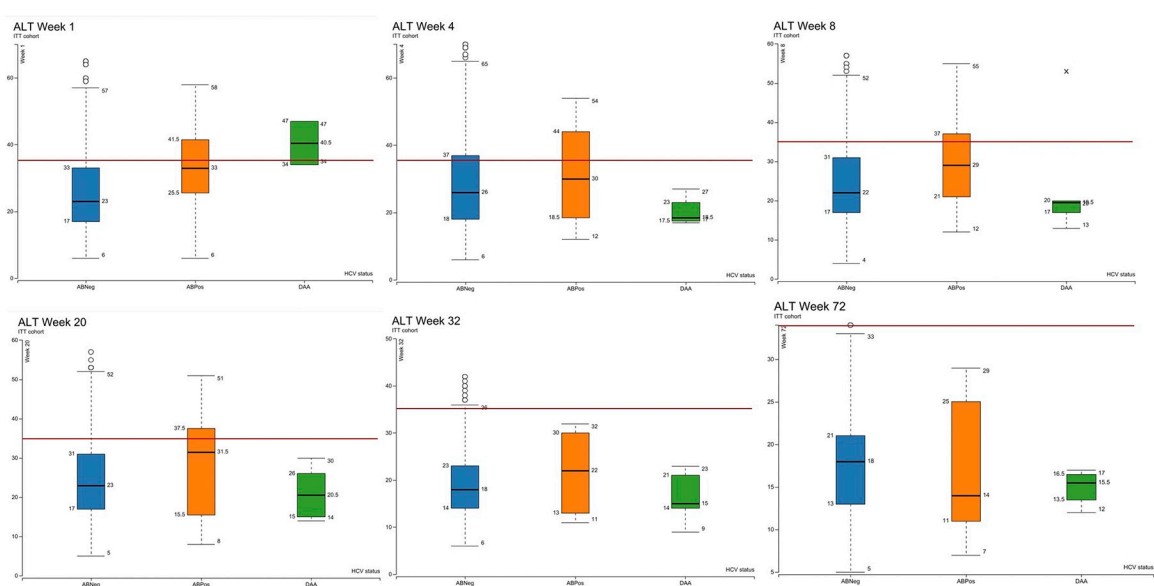

**Fig 3. Showing alanine aminotransferase (ALT) levels (IU/L) measured at weeks 1, 4, 8, 20, 32 and 72 of the study by hepatitis C status ABNeg: Hepatitis C antibody negative, ABPos: Hepatitis C antibody positive, DAA: Hepatitis C antibody positive treated with direct acting antivirals.** Red line marks upper limit of normal (35 IU/L).

group AST then decreased as the study progressed, while the ABPos group remained high until week 32. The mean ALP and bilirubin results were within normal limits in all groups during the study period.

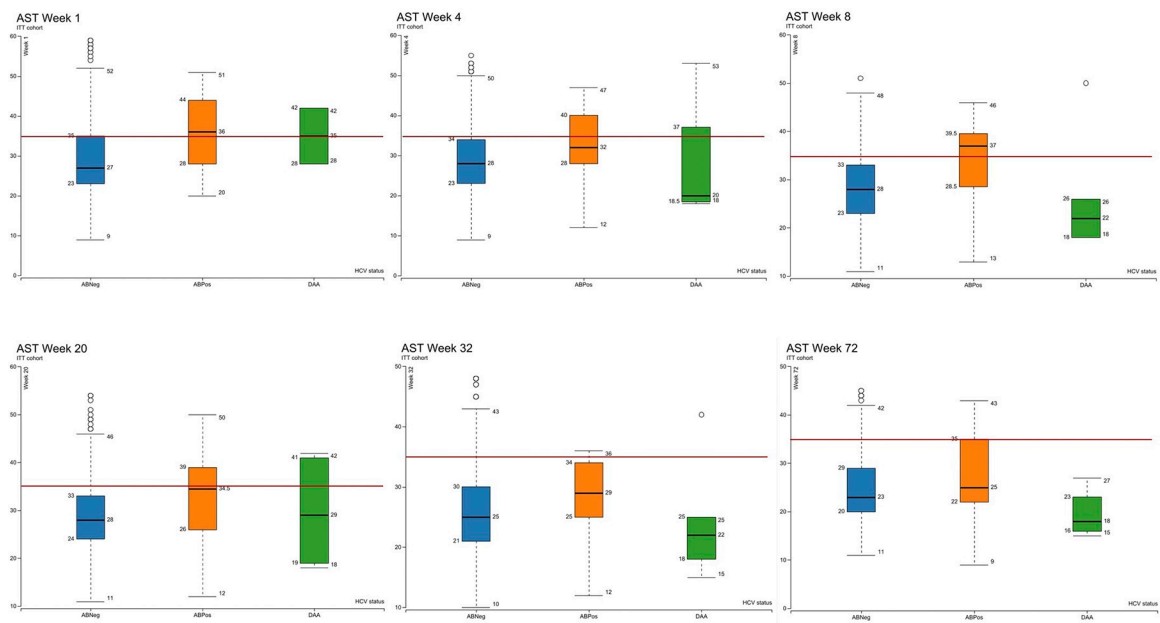

**Fig 4. Showing aspartate aminotransferase (AST) levels (IU/L) measured at weeks 1, 4, 8, 20, 32 and 72 of the study by hepatitis C status ABNeg: Hepatitis C antibody negative, ABPos: Hepatitis C antibody positive, DAA: Hepatitis C antibody positive treated with direct acting antivirals.** Red line marks upper limit of normal (35 IU/L).

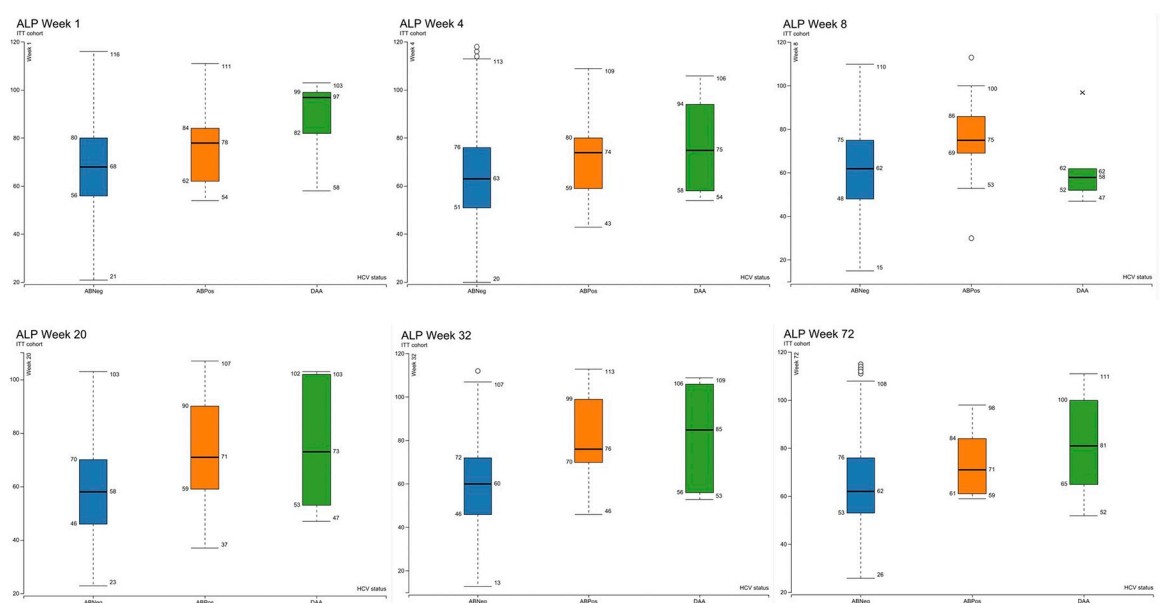

**Fig 5. Showing alkaline phosphatase (ALP) levels (IU/L) measured at weeks 1, 4, 8, 20, 32 and 72 of the study by hepatitis C status ABNeg: Hepatitis C antibody negative, ABPos: Hepatitis C antibody positive, DAA: Hepatitis C antibody positive treated with direct acting antivirals.** Upper limit of normal (130IU/L) not shown, as no values reached this threshold.

## Discussion

In TB-PRACTECAL, a small cohort of patients were concurrently treated for MDR-TB and HCV and the treatment outcomes were satisfactory for both conditions. There is some evidence that treatment with DAAs improved liver function tests. For those testing positive for

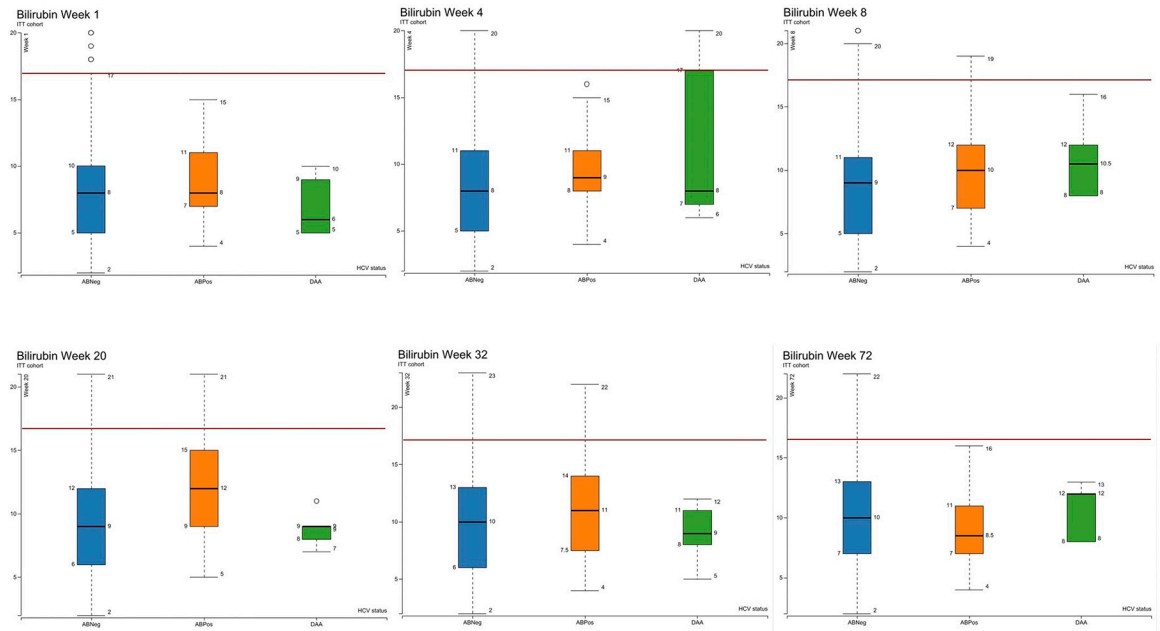

**Fig 6. Showing bilirubin levels (μmol/L) measured at weeks 1, 4, 8, 20, 32 and 72 of the study by hepatitis C status ABNeg: Hepatitis C antibody negative, ABPos: Hepatitis C antibody positive, DAA: Hepatitis C antibody positive treated with direct acting antivirals.** Red line marks upper limit of normal (17 μmol/L).

HCV, we noted that the ABPos group had a higher proportion of unfavourable TB outcomes than the ABneg group (44 vs 22%).

In TB-PRACTECAL seroprevalence for HCV was 4%, which, whilst noteworthy, is less than other similar studies across multi-country sites [4–6]. This is likely due to the South African site representing a large proportion of participants in the trial where the prevalence of chronic HCV is known to be lower than in Uzbekistan and Belarus [4]. In the small cohort of patients concurrently treated for MDR TB and HCV, none had significant gastrointestinal, haematological, or hepatic adverse events related to HCV treatment. This provides evidence that concurrent treatment with DAA for HCV and MDR TB with bedaquiline and pretomanid containing regimens may be safe. The evidence of improved liver function tests in the DAA group from baseline potentially suggests that reductions in transaminases can be achieved with concurrent treatment. This was also true for the 4 patients who were co-infected with MDR-TB, HCV, and HIV. The achievement of good clinical outcomes in this cohort builds on the existing evidence around the feasibility of concomitant treatment [17–20].

The difference of unfavourable TB outcomes in the ABpos and ABneg group (44 vs 22%) could be explained by the possibility of some participants having active HCV, this suggests that concurrently treating the HCV could improve MDR-TB treatment outcomes, however this requires further study in larger groups. This would support evidence from Tunesi et al who reported an improvement in liver dysfunction after 4 weeks of DAA treatment, suggesting HCV treatment may be protective in the face of co-infection [20]. Another possible explanation for the difference in outcomes could be co-infection with HIV and Hepatitis B in the ABPos group. The time to starting DAAs was earlier on TB-PRACTECAL with comparative outcomes, suggesting that clinicians could consider starting DAAs early as they might limit additional liver dysfunction and improve outcomes.

In Armenia, where concomitant treatment for HCV and MDR/RR-TB has been piloted prior to recent MDR/RR-TB treatment recommendation changes, feasibility studies found that integrating care was acceptable to both healthcare providers and patients. Some patients reported that they did not appreciate the higher daily pill burden, highly relevant in populations with higher HIV prevalence. The outcomes of TB-PRACTECAL mean a higher pill burden may be less of a concern for future patients concomitantly treated as the regimens in the trial had far fewer pills than the standard of care used in Armenia at the time of the study [24]. More data on this is invaluable as the close support offered during drug resistant TB treatment lends itself towards a model of concomitant HCV and TB care [24].

Limitations include small sample size, retrospective data collection and lack of randomization. Due to the lack of confirmatory HCV PCR results in the ABPos cohort, the proportion with chronic active hepatitis was unknown, however access to DAA treatment in all participating countries until recently has been limited making prior treatment unlikely. Spontaneous clearance is expected in around 20–25% of those infected. In this observational study, chronic HCV was treated at the discretion of the investigator, it is possible participants with better baseline clinical statuses were selected for treatment leading to better outcomes that what may be found in a randomized clinical trial.

## Conclusions

This small cohort of HCV positive patients provides additional evidence that concomitant treatment of MDR-TB with bedaquiline and pretomanid containing regimens and HCV infection with DAAs is feasible. Further data is required to confirm the safety and efficacy of concomitant treatment. Using existing TB frameworks to screen for and treat both diseases could represent an important opportunity to potentially impact on both epidemics.

## Supporting information

**S1 Checklist. Evidence that requirements for submission of cohort study.**
(DOC)

## Author Contributions

**Conceptualization:** Jocelyn Jansen van Vuuren, Tim Crocker-Buque, Catherine Berry.

**Data curation:** Jocelyn Jansen van Vuuren, Tim Crocker-Buque, Dzmitry Viatushka, Ruzilya Usmanova, Nargiza Parpieva, Irina Liverko, Varvara Solodovnikova.

**Formal analysis:** Tim Crocker-Buque.

**Funding acquisition:** Catherine Berry.

**Project administration:** Jocelyn Jansen van Vuuren.

**Software:** Tim Crocker-Buque.

**Supervision:** Catherine Berry, Bern-Thomas Nyang'wa, Animesh Sinha.

**Validation:** Catherine Berry.

**Writing – original draft:** Jocelyn Jansen van Vuuren.

**Writing – review & editing:** Jocelyn Jansen van Vuuren, Tim Crocker-Buque, Catherine Berry, Ruzilya Usmanova.

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
