## [Decision Letter · Decision Letter 0]

23 May 2024

PGPH-D-24-00510

Prevalence, treatment, and outcomes of hepatitis C in an MDR/RR-TB trial cohort

Dear Dr. Jansen van Vuuren,

Thank you for submitting your manuscript to PLOS Global Public Health. After careful consideration, we feel that it has merit but does not fully meet PLOS Global Public Health’s publication criteria as it currently stands. Therefore, we invite you to submit a revised version of the manuscript that addresses the points raised during the review process.

We look forward to receiving your revised manuscript.

Kind regards,

N. Sarita Shah

Academic Editor

Journal Requirements:

Additional Editor Comments (if provided):

Reviewers' comments:

Reviewer's Responses to Questions

**Comments to the Author**

1. Does this manuscript meet PLOS Global Public Health’s publication criteria? Is the manuscript technically sound, and do the data support the conclusions? The manuscript must describe methodologically and ethically rigorous research with conclusions that are appropriately drawn based on the data presented.

Reviewer #1: Yes

Reviewer #2: Yes

2. Has the statistical analysis been performed appropriately and rigorously?

Reviewer #1: Yes

Reviewer #2: Yes

3. Have the authors made all data underlying the findings in their manuscript fully available (please refer to the Data Availability Statement at the start of the manuscript PDF file)?

Reviewer #1: Yes

Reviewer #2: No

4. Is the manuscript presented in an intelligible fashion and written in standard English?

Reviewer #1: Yes

Reviewer #2: Yes

5. Review Comments to the Author

Reviewer #1: Tuberculosis (TB) and hepatitis C are major public health challenges globally. There is currently little evidence to support recommendations for co-treatment of hepatitis C and MDR/RR TB. In this manuscript, the authors leveraged the TB-PRACTECAL trial data to examine (1) the prevalence of hepatitis C virus infection in trial participants and (2) outcomes for those who received concurrent treatment of MDR/RR TB and hepatitis C.

Patients received BPaL(M/C) or the local standard of care MDR/RR TB regimen. Those with confirmed Hepatitis C were treated with direct acting antivirals (DAAs) at the discretion of the study clinicians based on local guidelines. TB treatment outcome definitions were the same as those used in the TB-PRACTECAL trial.

24 of the 552 participants in the TB-PRACTECAL trial were HCV seropositive. Six of the 24 HCV seropositive participants received concurrent treatment for HCV and MDR/RR TB. The results showed that three participants were cured of HCV and three had no confirmatory HCV RNA, five completed TB treatment and one discontinued treatment due to a severe adverse reaction. Most adverse events, including elevated liver enzymes, were grade 1-2 and self-limiting.

Despite the small sample size and the lack of HCV RNA testing results in three of the concurrently treated participants, this analysis contributes supportive evidence that concurrent treatment of Hepatitis C and MDR/RR TB is well tolerated and results in favorable outcomes for patients.

My recommendation is to accept the manuscript with the following revisions:

Abstract, Line 29: I suggest adding “with direct acting antivirals (DAAs)” after hepatitis C to define DAA before using it in the results section of the abstract.

Abstract, Lines 33-34: The sentence “Participants who were HCV seronegative, seropositive and…” is confusing and should be revised. I suggest the following revision for clarity: “Unfavourable TB treatment outcomes were noted in 106/523 (22%) of the HCV-negative, 8/18 (44%) of the HCV-seropositive, and 2/7 (29%) of participants treated with DAAs.”

Abstract, Lines 34-36: Consider revising this sentence to the following: “Of the six participants who received concurrent HCV and MDR/RR TB treatment, three were cured of HCV and three had no confirmatory HCV RNA, five completed TB treatment and one discontinued treatment due to a severe adverse reaction.”

Lines 90-93: I suggest revising this sentence to: “Included variables were demographics, co-morbidities (hepatitis B and HIV serostatus), and social history (alcohol and drug use), TB treatment drugs, biochemical tests (haematological and liver function test results), adverse event reports, clinical trial outcomes, and, where applicable, HCV treatment outcomes.”

Line 98: I suggest replacing “full” with “complete.”

Line 104: I suggest replacing “derangement” with “elevation.”

Line 190: I suggest replacing “reduced” with “declined” or “decreased.”

Lines 191-192: I suggest revising the last sentence to the following: “The mean ALP and bilirubin results were within normal limits in all groups during the study period.”

Line 194: I suggest replacing “significant” with “noteworthy.”

Lines 206-209: The authors discuss how the ABPos group had a higher proportion of unfavorable TB treatment outcomes compared to the ABNeg group. While treatment of concurrent active hepatitis C is an important consideration, the authors should also consider that the ABPos group could have had worse outcomes because of a higher proportion of Hepatitis B and HIV.

Line 212: I suggest revising “less” to “fewer.”

Reviewer #2: The article presents data from a TB-PRACTECAL clinical trial. The authors conducted a secondary data analysis to describe the co-treatment of HCV infection and MDR/RR TB. Concurrent treatment of MDR-TB and HCV showed promising results and a satisfactory profile of adverse events. Despite the small sample size of co-treated patients, this study provides an important piece of evidence and supports the benefits of concomitant treatment of hepatitis C and MDR/RR TB. Below, I am providing several suggestions to improve the manuscript.

Introduction

Line 36: The term “Confirmatory” is usually used in the context of confirming the initial diagnosis of HCV infection. If the authors mean the RNA test at the end of treatment, I would suggest to use different wording (e.g., end-of-treatment RNA test or something similar)

Line 52: When authors mention the WHO guidelines for MDR TB treatment, a reference should be added to the mentioned guideline.

Lines 52-53: The information provided in lines 52-53 is outdated and based on the older reference. I recommend using more recent estimates from the WHO. For example, Global progress report on HIV, viral hepatitis and sexually transmitted infections, 2021.

Line 67: The authors state that the aim was to examine the outcomes of HCV seropositive participants who were concomitantly treated. For HCV infection treatment, it would be required that the patients are viremic, not only seropositive. I recommend refining the language to clarify this.

Methods

Line 83: Please clarify if viremia results were available and, if not, why. It is understandable that patients would only be treated if they had viremia, but it is not clear from the text and might confuse some readers into thinking that some patients were treated based on antibody results. Below it is mentioned that PCR testing was conducted, but it is unclear why that is not reported.

Line 104: Liver function tests are later mentioned as acronyms. Please define the acronyms here at the first mention.

Line 111: It is unclear how the participants were included in the trial to start with if they had unconfirmed MDR/RR-TB. I would recommend clarifying in the first paragraph of the methods section the eligibility criteria for including patients in the trial.

Line 116: Typo, the word “use” is repeated.

Results

Line 154: It seems like the sentence “Mean ALT measured at screening was slightly higher (36 IU/L) and above the upper limit of normal range…” refers to the DAA group, which should be clarified.

Line 172: Please clarify if the HCV viral load in three participants was measured at the time of completion or after 12 weeks to check for sustained virologic response?

Line 189-190: From the sentence, it is unclear if AST was at the upper limit of normal in all participants in the DAA group or on average.

Discussion

Line 194: I recommend starting the discussion section with a summary of the main findings before moving to the seropositivity prevalence discrepancies between sites.

Line 224: Related to my comment above, please explain either here or in the methods why PCR results were not available -

Line 223-227: Another potential limitation is confounding by indication due to the study's observational nature and lack of randomization. In the methods it was mentioned that treatment for HCV infection was at the discretion of the investigator. It is possible that doctors prescribe concomitant HCV treatment only to patients with better baseline clinical conditions, and this kind of study might find better clinical outcomes and safety profiles than what we would observe if a randomized clinical trial were conducted.

Figures

The box plots provided in the supplementary material are very interesting, and I consider them highlights of the results from this study. If possible, I recommend including them in the main paper.

6. PLOS authors have the option to publish the peer review history of their article (what does this mean?). If published, this will include your full peer review and any attached files.

**Do you want your identity to be public for this peer review?** For information about this choice, including consent withdrawal, please see our Privacy Policy.

Reviewer #1: No

Reviewer #2: No

---

## [Editor Report · Decision Letter 1]

19 Aug 2024

Prevalence, treatment, and outcomes of hepatitis C in an MDR/RR-TB trial cohort

PGPH-D-24-00510R1

Dear Dr Jansen van Vuuren,

We are pleased to inform you that your manuscript 'Prevalence, treatment, and outcomes of hepatitis C in an MDR/RR-TB trial cohort' has been provisionally accepted for publication in PLOS Global Public Health.

Best regards,

N. Sarita Shah

Academic Editor